# Active Alignment of Large-Aperture Space Telescopes for Optimal Ellipticity Performance

**DOI:** 10.3390/s23104705

**Published:** 2023-05-12

**Authors:** Xiaoquan Bai, Xixi Gu, Boqian Xu, Fengyi Jiang, Zhirong Lu, Shuyan Xu, Guohao Ju

**Affiliations:** 1Changchun Institute of Optics, Fine Mechanics and Physics, Chinese Academy of Sciences, Changchun 130033, China; baixiaoquan@ciomp.ac.cn (X.B.); guxixi@ciomp.ac.cn (X.G.); ciomp_xubq@126.com (B.X.); jiangfengyi@ciomp.ac.cn (F.J.); luzhirong18@mails.ucas.edu.cn (Z.L.); xusy@ciomp.ac.cn (S.X.); 2Chinese Academy of Sciences Key Laboratory of On-Orbit Manufacturing and Integration for Space Optics System, Changchun 130033, China

**Keywords:** active optical alignment, ellipticity performance, space telescope

## Abstract

Ellipticity performance of space telescopes is important for exploration of dark matter. However, traditional on-orbit active optical alignment of space telescopes often takes “minimum wavefront error across the field of view” as the correction goal, and the ellipticity performance after correcting the wave aberration is not optimal. This paper proposes an active optical alignment strategy to achieve optimal ellipticity performance. Based on the framework of nodal aberration theory (NAT), the aberration field distribution corresponding to the optimal full field-of-view ellipticity is determined using global optimization. The degrees of freedom (DOFs) of the secondary mirror and the folded flat mirror are taken as the compensation DOFs to achieve the optimal ellipticity performance. Some valuable insights into aberration field characteristics corresponding to optimal ellipticity performance are presented. This work lays a basis for the correction of ellipticity for complicated optical systems.

## 1. Introduction

Point-spread function (PSF) ellipticity of space telescopes is of great significance for the exploration of dark matter and dark energy. Dark matter and dark energy do not absorb, emit or radiate light, so they cannot be observed directly, and can only be observed indirectly by using the weak gravitational lensing effect. At present, the strength of the weak gravitational lensing effect is mainly measured by the change in the ellipticity of stars or galaxies. Therefore, accurate measurement of the ellipticity of stars or galaxies in the universe is of great significance for the observation and research of dark matter and dark energy.

Influence of weak gravitational lens effect on the ellipticity of celestial bodies or galaxies is very weak [1,2,3]. In order to accurately measure it, the change in ellipticity caused by the optical system itself should be limited to a very small range so as to minimize the influence of the ellipticity of the optical system on the observation of dark matter and dark energy. The ellipticity of the optical system itself is mainly caused by the asymmetrical pupil shape and wave aberration of the optical system itself [4]. Generally speaking, the smaller the wave aberration is, the smaller the ellipticity of the point-spread function is. However, the “full field wave aberration optimization” and “full field point-spread function ellipticity optimization” of optical system essentially correspond to two different system states and cannot be confused.

It is critical to align large aperture space telescope on orbit [5,6]. At present, active alignment methods are mainly divided into two classes: numerical method and analytic method. On the numerical computing side, sensitive table method (STM) [7], artificial neural networks method (ANN) [8], merit function (MF) regression method [9,10,11], differential wavefront sampling (DWS) [12,13,14] and image features [15,16,17] are widely used in ground alignment. The analytical method is mainly nodal aberration theory (NAT) [18,19,20,21,22,23]. Importantly, NAT provides an aberration model that yields to obtain insight into the aberration response to misalignments [24,25,26,27] and guide optical alignment [28,29,30,31,32,33,34,35,36,37].

However, traditional on-orbit active optical alignment of space telescopes often takes “optimal wavefront aberration in the full field of view” as the correction goal, and the ellipticity performance after correcting the wave aberration is not optimal. It is necessary to formulate different correction targets for different astronomical observation needs of space telescopes. Specifically, when observing dark matter and dark energy, the mode of “optimal ellipticity performance in the full field of view” is adopted to make the point-spread function ellipticity caused by the optical system itself as small as possible so as to better identify and judge the weak gravitational lens effect caused by dark matter and dark energy.

In this paper, an active alignment strategy to achieve optimal ellipticity performance is proposed. Chinese Space Station Telescope (CSST) is taken as an example to demonstrate its performance. (At the beginning of CSST design, the distribution of ellipticity and wavefront aberration was balanced. Correcting ellipticity is one of the requirements for CSST in orbit. The results of integrated simulation analysis (for CSST) are used for simulation analysis in this paper.). Based on the framework of nodal aberration theory (NAT), the aberration field distribution corresponding to the optimal full field-of-view ellipticity is determined using global optimization. In this process, the objective function is the mean ellipticity value of 41*41 field points and all misalignments of mirrors are taken as the variable. Then, the degrees of freedom (DOFs) of the secondary mirror and the folded flat mirror are taken as the compensation DOFs to achieve the optimal ellipticity performance on orbit. Some valuable insights into aberration field characteristics corresponding to the optimal ellipticity performance are also presented concerning the proportion of different aberration types.

## 2. Ellipticity

The ellipticity of PSF under certain perturbations is defined as follows [38,39,40,41,42,43]:(1)$$e\left(\vec{H},v\right)=\sqrt{{\left[e_{1}\left(\vec{H},v\right)\right]}^{2}+{\left[e_{2}\left(\vec{H},v\right)\right]}^{2}},$$
where v includes misalignment parameters of the optical system, $\vec{H}$ represents field position of the PSF, and $e_{1}$ and $e_{2}$ are the two components of the PSF ellipticity, which can be expressed as
(2)$$\left\{\begin{array}{l} e_{1}\left(\vec{H},v\right)=\frac{Q_{XX}\left(\vec{H},v\right)-Q_{YY}\left(\vec{H},v\right)}{Q_{XX}\left(\vec{H},v\right)+Q_{YY}\left(\vec{H},v\right)}\\ e_{2}\left(\vec{H},v\right)=\frac{2Q_{XY}\left(\vec{H},v\right)}{Q_{XX}\left(\vec{H},v\right)+Q_{YY}\left(\vec{H},v\right)} \end{array}\right.,$$
where $Q_{XX}$, $Q_{YY}$ and $Q_{XY}$ are the second-order moments of the PSF,
(3)$$\left\{\begin{array}{l} Q_{XX}\left(\vec{H},v\right)=\frac{\int I\left(\vec{H},v;x,y\right)w\left(x,y\right){\left(x-\bar{x}\right)}^{2}dxdy}{\int I\left(\vec{H},v;x,y\right)w\left(x,y\right)dxdy}\\ Q_{XY}\left(\vec{H},v\right)=\frac{\int I\left(\vec{H},v;x,y\right)w\left(x,y\right)\left(x-\bar{x}\right)\left(y-\bar{y}\right)dxdy}{\int I\left(\vec{H},v;x,y\right)w\left(x,y\right)dxdy}\\ Q_{YY}\left(\vec{H},v\right)=\frac{\int I\left(\vec{H},v;x,y\right)w\left(x,y\right){\left(y-\bar{y}\right)}^{2}dxdy}{\int I\left(\vec{H},v;x,y\right)w\left(x,y\right)dxdy} \end{array}\right.,$$
where I represents the two-dimensional intensity distribution of the PSF at certain field position with certain misalignment parameters, x and y are coordinates in the PSF image, $\bar{x}$ and $\bar{y}$ represent the centroid position of the PSF image, and w is a matrix representing Gaussian weighting coefficient.

The intensity distribution of a PSF at certain field position with certain misalignment parameters can be expressed according to Fourier optics and nodal aberration theory, which can be expressed as [15]
(4)$$I\left(\vec{H},v\right)={\left\{FT^{-1}\left\{P\right.exp\left[i\frac{2\pi }{\lambda }\left(W\left(\vec{H},v\right)\right)\right]\right\}}^{2},$$
where i is an imaginary unit, $FT^{-1}$ is the inverse Fourier transform, and P is a two-dimensional matrix representing the intensity distribution of the exit pupil surface (the element value of the matrix is 1 in the normalized aperture, and the rest are 0). $W\left(\vec{H},v\right)$ represents the wavefront phase at field $\vec{H}$ with system misalignment parameters v. Importantly, based on the framework of nodal aberration theory, $W\left(\vec{H},v\right)$ can be expressed as the sum of two parts,
(5)$$W\left(\vec{H},v\right)=W_{0}\left(\vec{H}\right)+W_{M}\left(\vec{H},v\right),$$
where $W_{0}\left(\vec{H}\right)$ represents the aberration field in the original state which is determined by the optical design and mirror figures (unaffected by misalignments), and $W_{M}\left(\vec{H},v\right)$ represents the net aberration field induced by misalignments, which can be expressed as a linear function of v and $\vec{H}$. The specific expression of $W_{M}\left(\vec{H},v\right)$ can be referred to [15].

## 3. Active Alignment for Optimal Ellipticity Performance

Using Equations (1)–(5), the relationships between the PSF ellipticity at arbitrary field position and the misalignment parameters are established. On this basis, we can set average ellipticity across the field of view as objection function
(6)$$E\left(v\right)=\frac{\sum_{k=1}^{K}\sqrt{{\left[e_{1}\left({\vec{H}}_{k},v\right)\right]}^{2}+{\left[e_{2}\left({\vec{H}}_{k},v\right)\right]}^{2}}}{K},$$
where K represents the number of sampling points in the field of view, and the specific value of it depends on the computing power and the calculation accuracy of the average ellipticity (for example, K is 11*11 in this paper, etc.). Particle swarm optimization algorithm is used in this paper. Gradient-based optimization algorithms can easily be trapped in a local minimum, for the searching direction of them mainly depends on the derivative information of the error metric. At the same time, the objective function in this paper does not have an advantage in taking derivatives. From this perspective, PSO algorithms are better, for they rely directly upon objective function values rather than derivative information [44]. We set the learning factors c1 = 1.6, c2 = 2.65. The value of w is constant, w = 0.5, T = 0.001. The population size is 40 and the maximum number of iterations is 800. Using the numerical optimization tool, the set of misalignment parameter ($v_{0}$) corresponding to the minimum average ellipticity can be obtained:(7)$$E\left(v_{0}\right)=min\left\{E\left(v\right)\right\}.$$

We can then obtain the aberration field corresponding to optimal ellipticity performance.

Note that in the process presented above, all the misalignment parameters are considered. However, when correcting aberration on orbit, only part of the parameters can be selected as compensating DOFs due to constraints in space. For large-aperture off-axis TMA telescopes with freeform surface, secondary mirror (SM) and fine steering mirror (FSM) can be selected as the compensating DOFs based on aberration compensation characteristics. Then, the aberration field corresponding to the optimal ellipticity performance is taken as the correction goal; the DOFs of the SM and FSM are utilized to achieve this goal.

The diagram of the proposed approach is illustrated in Figure 1. This approach can be divided into the following two parts:(1)In the first part, we obtain the aberration field corresponding to the optimal ellipticity performance through optimization. The mean ellipticity value of 11*11 field points is taken as the objective function and the misalignments of each mirror are taken as the variable. Meanwhile, the relationships between the PSF ellipticity at arbitrary field position and the misalignment parameters are established based on Fourier optics and nodal aberration theory.(2)In the second part, SM and the FSM are selected as the adjustable degrees of freedom for the optimal correction of ellipticity, and the adjustment amount of SM and FSM is calculated (multi-field wavefront aberrations are needed in advance).

In Figure 1, $W\left(\vec{H},v_{0}\right)$ represents the aberration fields corresponding to the optimal ellipticity performance obtained in the first step using global optimization, $W\left(\vec{H},v_{1}^{\prime }\right)$ represents the aberration fields after adjusting SM and FSM on orbit to achieve the optimal ellipticity performance.

## 4. Simulations

### 4.1. Optical System Used for Demonstrating the Proposed Active Alignment Strategy

In this paper, the optical system of the Chinese Space Station Telescope (CSST) is used to demonstrate the effectiveness of the proposed approach, which is a 2 m F/14 COOK-type unobscured off-axis three-mirror anastigmatic (TMA) telescope with 1.1° × 1° field of view (FOV). The optical layout of the CSST is shown in Figure 2, including primary mirror (PM), SM, tertiary mirror™, which is a freeform surface, a folding mirror (FM), which is used to correct image plane tilt and switch back-end astronomical measuring instruments, and an image plane (IMG) [45,46]. The materials used to manufacture mirrors of CSST are SiC [47,48,49,50].

To maintain the exceptional image quality of the CSST, wavefront sensing and control (WFS and C) is vital to ensure the optical alignment of the telescope throughout the mission. For CSST, the five degrees of freedom (DOFs) of SM (*XDE_SM_*, *YDE_SM_*, *ADE_SM_*, *BDE_SM_* and *ZDE_SM_*) and three DOFs of the FM (*ADE_FM_*, *BDE_FM_* and *ZDE_FM_*) are selected as the adjustable DOFs to align and compensate for the effects of misalignments on wavefront aberrations (*XDE* and *YDE* are the mirror vertex decenters in the x-z and y-z plane, respectively, *BDE* and *ADE* are the mirror tip-tilts in the x-z and y-z plane, respectively, and *ZDE* represents the errors in spacing between mirrors). Specifically, SM is used to compensate for the non-rotationally symmetric aberrations of the PM and TM. FM is used to correct defocused aberration and inclination of the image plane.

To evaluate the ellipticity performance of the CSST, 41 × 41 PSFs are generated by the CODE V corresponding to 41 × 41 field points uniformly distributed in the FOV. The PSF is sampled on a 512 × 512 grid with an image sampling step of 0.4 micron [4] in this section. In addition, the root mean square wavefront errors (RMS WFE) of these 41 × 41 field points are used to evaluate the wavefront aberration of the CSST. The full-field distribution of ellipticity and RMS WFE distribution in nominal state are shown in Figure 3a and Figure 3b, respectively. The maximum and mean values are 0.0843 and 0.0234, as shown in Figure 3a. The maximum and mean values are 0.0523λ and 0.0256λ, as shown in Figure 3b.

However, the mirror has certain surface deformations due to manufacturing error, gravity loading and other reasons. The existence of these deformations reduces the imaging quality of the system, which can also lead to changes in the ellipticity distribution of the system. According to the characteristics of the support structure of the CSST, the simulation group provides the surface deformation caused by changes in external environmental factors such as gravity unloading and temperature variations. These surface deformations can be loaded into the optical model through the CODE V software. The figures of each mirror (including surface deformation) considered in this paper are shown in Figure 4.

The full-field distribution of ellipticity and the full-field distribution of RMS WFE distribution in the presence of surface deformation are shown in Figure 5a and Figure 5b, respectively. The maximum and mean values are 0.1152 and 0.0351, as shown in Figure 5a. The maximum and mean values are 0.0778λ and 0.0472λ, as shown in Figure 5b. Comparing Figure 5 and Figure 3, we can see that after introducing the set of surface deformations, both full-field distribution of ellipticity and RMS WFE deviate from the nominal state.

### 4.2. Comparison of Ellipticity Distribution between the State of “Optimal Wavefront Aberration” and “Optimal Ellipticity Performance” in the Presence of Surface Deformation

The wavefront aberrations induced by surface deformations can be compensated by wavefront aberrations induced by misalignments to some extent (only the low-order wavefront aberrations induced by surface deformation can be compensated). In this section, full-field distribution of ellipticity and RMS WFE under different correction targets are compared in the presence of surface deformation.

On the one hand, “optimal wavefront aberration in the full FOV” is taken as the correction goal of on-orbit active optical alignment. Referring to JWST [5], the Sensitivity Table Method (STM) is selected to calculate the alignments of SM and FM for compensating wavefront aberrations. For the given set of surface deformations, the amount of adjustable DOFs required to correct the wavefront are *XDE_SM_* = −0.0367 mm, *YDE_SM_* = 0.0061 mm, *ZDE_SM_* = −0.0078 mm, *ADE_SM_* = 0.0006°, *BDE_SM_* = 0.0007°, *ZDE_FM_* = 0.1178 mm, *ADE_FM_* = 0.0019° and *BDE_FM_* = −0.0001°. The full-field distribution of ellipticity and RMS WFE after correction are shown in Figure 6a and Figure 6b, respectively. The maximum and mean values are 0.0857 and 0.0369 in Figure 6a. The maximum and mean values are 0.0647λ and 0.0426λ in Figure 6b.

On the other hand, “optimal ellipticity performance in the full FOV” described in Section 3 is taken as the correction goal of on-orbit active optical alignment. The amount of adjustable DOFs required to correct the ellipticity are *XDE_SM_* = −0.0662 mm, *YDE_SM_* = −0.0346 mm, *ZDE_SM_* = 0.0068 mm, *ADE_SM_* = −0.0013°, *BDE_SM_* = 0.0019°, *ZDE_FM_* = −0.0040 mm, *ADE_FM_* = 0.0003° and *BDE_FM_* = −0.0029°. The full-field distribution of ellipticity and RMS WFE after correction are shown in Figure 7a,b. The maximum and mean values are 0.0670 and 0.0228, as shown in Figure 7a. The maximum and mean values of RMS WFE are 0.0825λ and 0.0492λ, as shown in Figure 7b.

Comparing Figure 6 and Figure 7, we can recognize that a different correction goal will lead to a different ellipticity and RMS WFE performance, as shown in Table 1. Importantly, the mean value of ellipticity in the full FOV for the state of “optimal wavefront aberration” and the state of “optimal ellipticity performance” are 0.0369 and 0.0228, respectively. This fact demonstrates that the method proposed in this paper can effectively improve the ellipticity performance. Meanwhile, the mean value of RMS WFE in the full FOV for the state of “optimal wavefront aberration” and the state of “optimal ellipticity performance” are 0.0426 and 0.0492, respectively. The mean value of RMS WFE in the full FOV for the state of “optimal ellipticity performance” is also acceptable (which is still within the diffraction limit).

### 4.3. Comparison of Ellipticity Distribution between the State of “Optimal Wavefront Aberration” and “Optimal Ellipticity Performance” in the Presence of Surface Deformation and Mirror Misalignments

This subsection continues to discuss that the full-field distribution of ellipticity and RMS WFE under a different active alignment goal in the presence of both surface deformation and mirror misalignments (including lateral and axial misalignments [25,26,27]).

A set of random mirror misalignment is generated within the predicted misalignment ranges, which are presented below: *XDE_PM_* = −0.0032 mm, *YDE_PM_* = 0.0020 mm, *ZDE_PM_* = 0.0036 mm, *ADE_PM_* = 0.0002°, *BDE_PM_* = 0.0009°, *XDE_SM_* = 0.0033 mm, *YDE_SM_* = −0.0001 mm, *ZDE_SM_* = −0.0042 mm, *ADE_SM_* = 0.0004°, *BDE_SM_* = 0.0009°, *XDE_TM_* = 0.0031 mm, *YDE_TM_* = −0.0041 mm, *ZDE_TM_* = 0.0033 mm, *ADE_TM_* = 0.0009°, *BDE_TM_* = −0.0003°, *XDE_FM_* = −0.0007 mm, *YDE_FM_* = −0.0042 mm, *ZDE_FM_* = −0.9935 mm, *ADE_FM_* = 0.0008° and *BDE_FM_* = 0.0001°. The full-field distribution of ellipticity and RMS WFE in the presence of both surface deformation and mirror misalignments are shown in Figure 8a and Figure 8b, respectively. The maximum and mean values are 0.1641 and 0.0854, as shown in Figure 8a. The maximum and mean values are 0.6881λ and 0.6448λ, as shown in Figure 8b.

Comparing Figure 8 and Figure 5, it can be seen that the influence of mirror misalignments on the full-field distribution of ellipticity and RMS WFE is more obvious than the influence of surface deformation. Therefore, it is of great significance to verify the correction effect of a different correction goal on the full-field distribution of ellipticity and RMS WFE in the presence of both surface deformation and mirror misalignments.

On the one hand, “optimal wavefront aberration in the full FOV” is taken as the correction goal of on-orbit active optical alignment. For the given set of and surface deformation and mirror misalignments, the amounts of adjustable DOFs required to correct the wavefront are *XDE_SM_* = 0.0497 mm, *YDE_SM_* = 0.0069 mm, *ZDE_SM_* = 0.0065 mm, *ADE_SM_* = 0.0006°, *BDE_SM_* = 0.0003°, *ZDE_FM_* = 1.0107 mm, *ADE_FM_* = −0.0011° and *BDE_FM_* = 0.0028°. The full-field distribution of ellipticity and RMS WFE after correction are shown in Figure 9a and Figure 9b, respectively. The maximum and mean values are 0.0810 and 0.0374, as shown in Figure 9a. The maximum and mean values are 0.0633λ and 0.0430λ, as shown in Figure 9b.

On the other hand, “optimal ellipticity performance in the full FOV” described in Section 3 is taken as the correction goal of on-orbit alignment. For the given set of surface deformation and mirror misalignments, the amounts of adjustable DOFs required to correct the ellipticity are *XDE_SM_* = 0.0202 mm, *YDE_SM_* = −0.0339 mm, *ZDE_SM_* = 0.0213 mm, *ADE_SM_* = −0.0013°, *BDE_SM_* = 0.0015°, *ZDE_FM_* = 0.8872 mm, *ADE_FM_* = −0.0028° and *BDE_FM_* = 0.0000°. The full-field distribution of ellipticity and RMS WFE after correction are shown in Figure 10a and Figure 10b, respectively. The maximum and mean values are 0.0664 and 0.0233, as shown in Figure 10a. The maximum and mean values are 0.0807λ and 0.0486λ, as shown in Figure 10b.

Comparing Figure 10 and Figure 9, we can further conclude that a different correction goal leads to a different ellipticity performance and RMS WFE performance, as shown in Table 2. From Figure 10 and Figure 7, it can be clearly seen that the method proposed by this paper has a certain correction effect on the ellipticity performance with the mean value of the RMS WFE under 1/14λ. On-orbit active alignment method with the goal of optimal ellipticity performance can effectively optimize the full-field ellipticity performance. While the RMS WFE in this state is not optimal, it is still comparable to the state of “optimal RMS WFE”. Therefore, the proposed method is specifically suitable for the case where high ellipticity performance is required in the presence of both surface deformation and mirror misalignments.

From Figure 6, Figure 7, Figure 9 and Figure 10, it can be seen that mirror misalignments do not change the optimal full-field distribution of ellipticity and RMS WFE within a certain range of misalignments, which is consistent with the aberration compensation theory. Some valuable insights into aberration field characteristics about on-orbit active alignment are discussed in the next section.

## 5. Results

Some valuable insights into aberration field characteristics corresponding to optimal ellipticity performance are presented in this section. The full-field distribution of defocus, astigmatism and coma in a different state are shown in Figure 11, Figure 12 and Figure 13.

The following conclusions can be drawn from Figure 11, Figure 12 and Figure 13:(1)The full-field distribution of defocus and astigmatism is the main factor affecting the full-field distribution of ellipticity, while the full-field distribution of coma has little influence on the full-field distribution of ellipticity.(2)At the same field point, different values of defocus and astigmatism can produce different values of ellipticity. On orbit, the full-field distribution of ellipticity can be corrected by adjusting the full-field distribution of defocus and astigmatism.(3)There exists an inherent relationship between the proportion of full-field distribution of defocus and astigmatism for the state of “optimal ellipticity performance”: if the value of defocused aberration at a certain field point is comparatively large, the value of astigmatism at the corresponding point is relatively small, and if the value of astigmatism at certain field point is large, the value of defocus at the corresponding field point is relatively small. However, there is no similar “relationship” when the correction goal is “optimal wavefront error”.

## 6. Conclusions

This paper proposes an active optical alignment strategy to achieve optimal ellipticity performance and demonstrates that there exists certain difference between the state of “optimal ellipticity performance” and the state of “optimal wavefront error”. Based on the framework of nodal aberration theory (NAT), an active alignment method of large-aperture space telescopes for optimal ellipticity performance is proposed, where the aberration field distribution corresponding to the optimal full field-of-view ellipticity is determined with global optimization, and the degrees of freedom (DOFs) of the secondary mirror and the folded flat mirror are taken as the compensation DOFs to achieve the optimal ellipticity performance. The simulations show that the mean ellipticity value corresponding to optimal ellipticity performance is 0.03, while this value is 0.04 for optimal wavefront performance (the optical system of CSST is used for simulation). Some valuable insights into aberration field characteristics corresponding to the optimal ellipticity performance are presented, which shows that the proportion between astigmatic aberration field and medial focal surface can affect the ellipticity performance. In practice, we can change the goal of on-orbit alignment from “optimal wavefront error” to the goal of “optimal ellipticity performance” if a higher ellipticity performance is required. This work lays a basis for the correction of ellipticity for complicated optical systems.

## Figures and Tables

**Figure 1 sensors-23-04705-f001:**
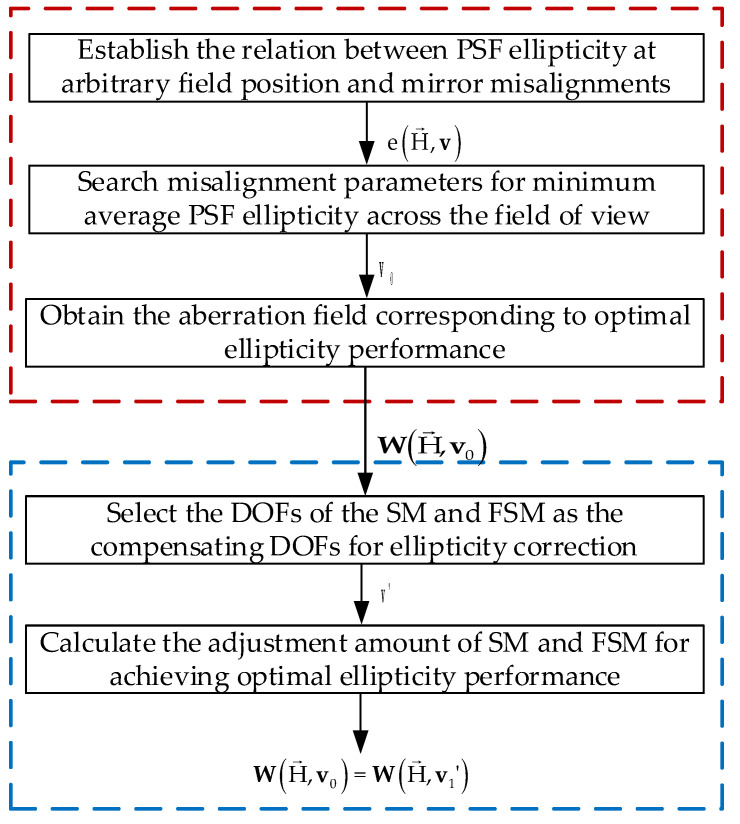
Diagram of the proposed on-orbit active alignment strategy for optimal ellipticity performance.

**Figure 2 sensors-23-04705-f002:**
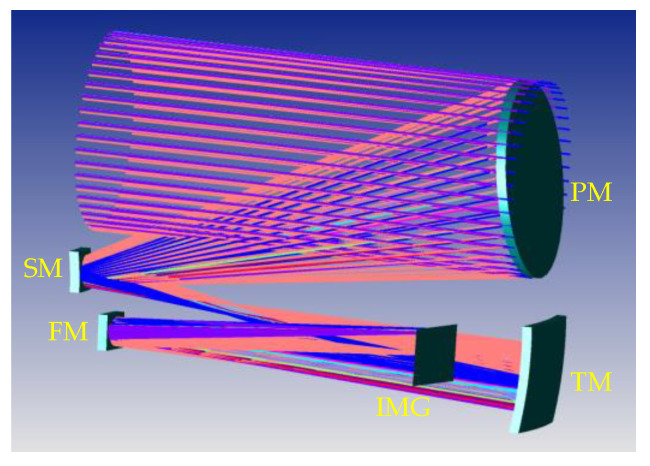
Optical layout of the CSST.

**Figure 3 sensors-23-04705-f003:**
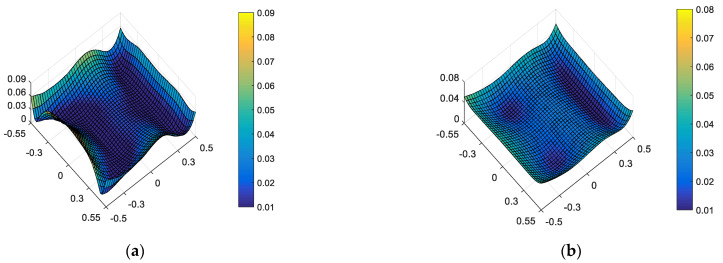
Full-field distribution of ellipticity (**a**) and RMS WFE (**b**) in nominal state.

**Figure 4 sensors-23-04705-f004:**
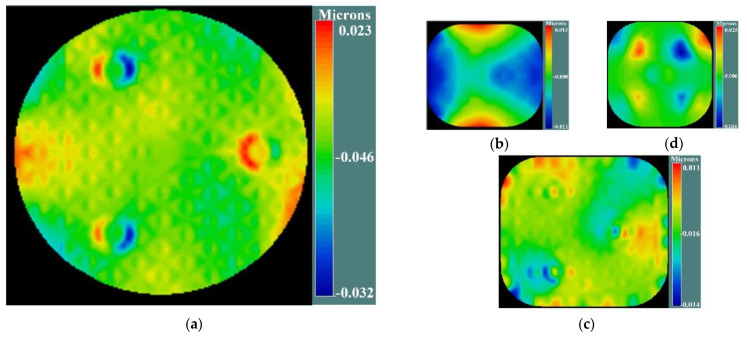
The figure of each mirror (including surface deformation) considered in this paper for CSST: (**a**) figure of PM, (**b**) figure of SM, (**c**) figure of TM, (**d**) figure of FM.

**Figure 5 sensors-23-04705-f005:**
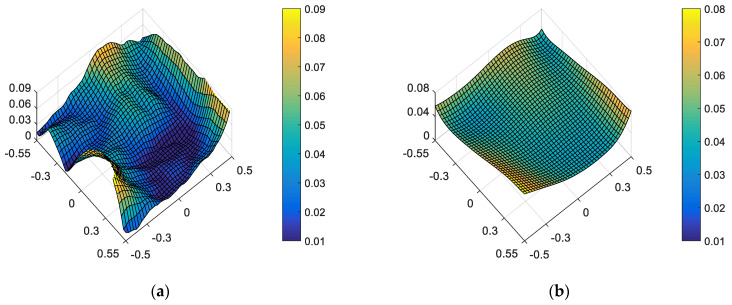
Full-field distribution of ellipticity (**a**) and RMS WFE (**b**) in the presence of surface deformations.

**Figure 6 sensors-23-04705-f006:**
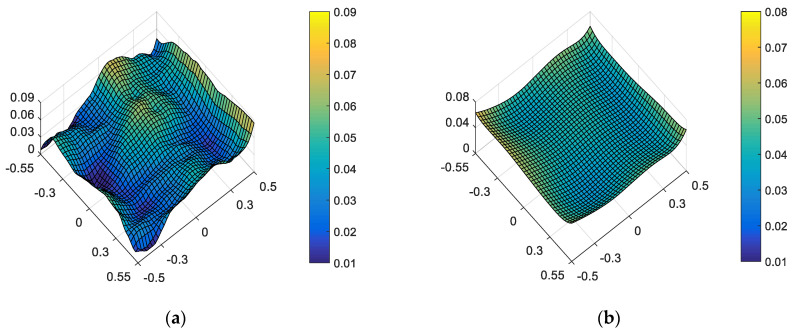
Full-field distribution of ellipticity (**a**) and RMS WFE (**b**) for the state of “optimal wavefront aberration”.

**Figure 7 sensors-23-04705-f007:**
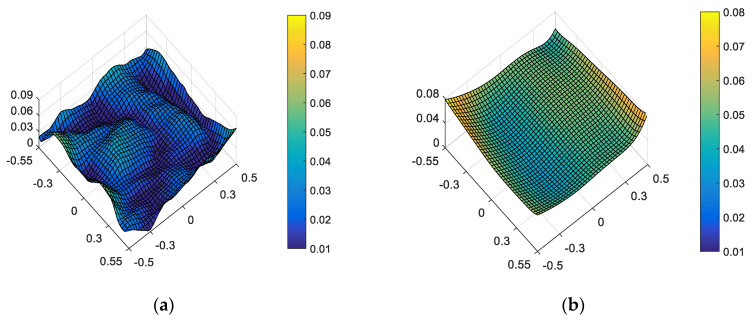
Full-field distribution of ellipticity (**a**) and RMS WFE (**b**) for the state of “optimal ellipticity performance”.

**Figure 8 sensors-23-04705-f008:**
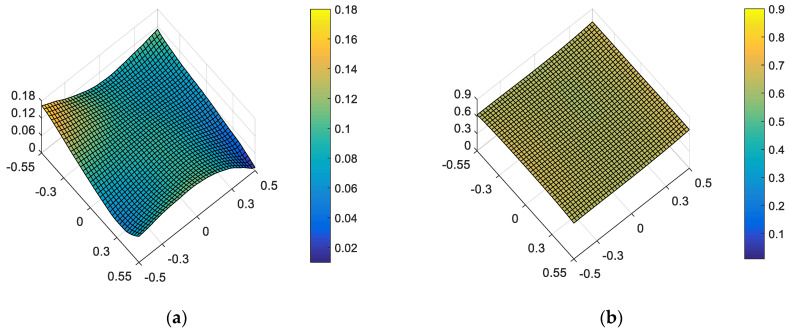
Full-field distribution of ellipticity (**a**) and RMS WFE (**b**) in the presence of surface deformation and mirror misalignments.

**Figure 9 sensors-23-04705-f009:**
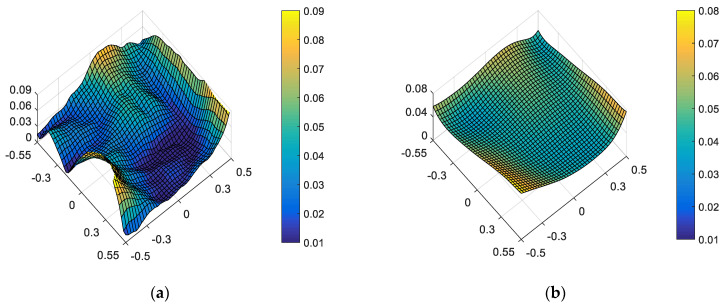
Full-field distribution of ellipticity and RMS WFE after optimizing by optimal wavefront aberration: (**a**) ellipticity distribution and (**b**) RMS WFE distribution.

**Figure 10 sensors-23-04705-f010:**
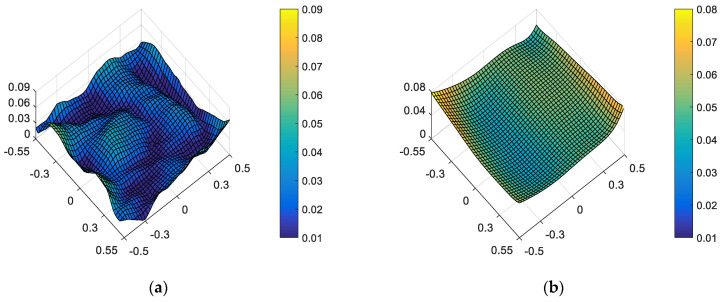
Full-field distribution of ellipticity and RMS WFE after optimizing by optimal ellipticity performance: (**a**) ellipticity distribution and (**b**) RMS WFE distribution.

**Figure 11 sensors-23-04705-f011:**
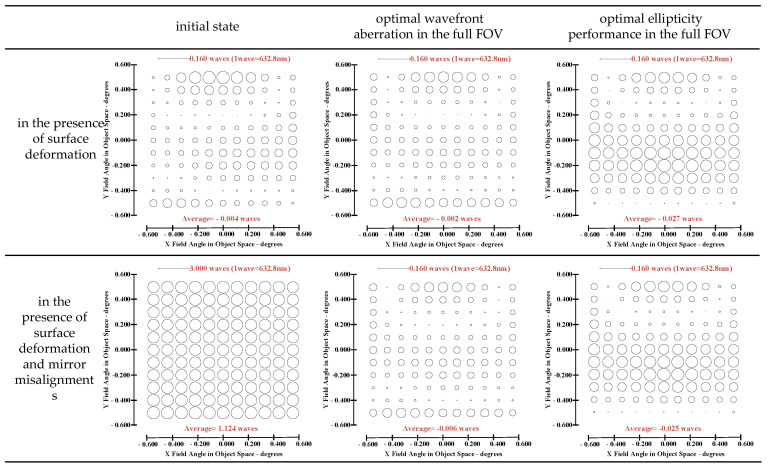
FFDs for defocus (Z4).

**Figure 12 sensors-23-04705-f012:**
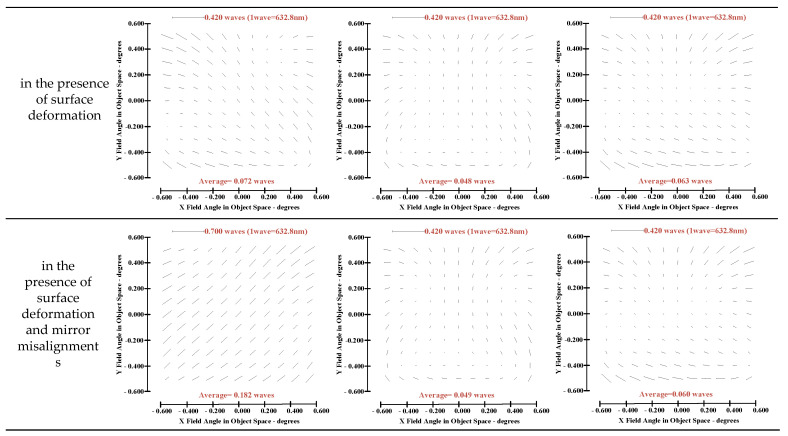
FFDs for astigmatism (Z5/Z6).

**Figure 13 sensors-23-04705-f013:**
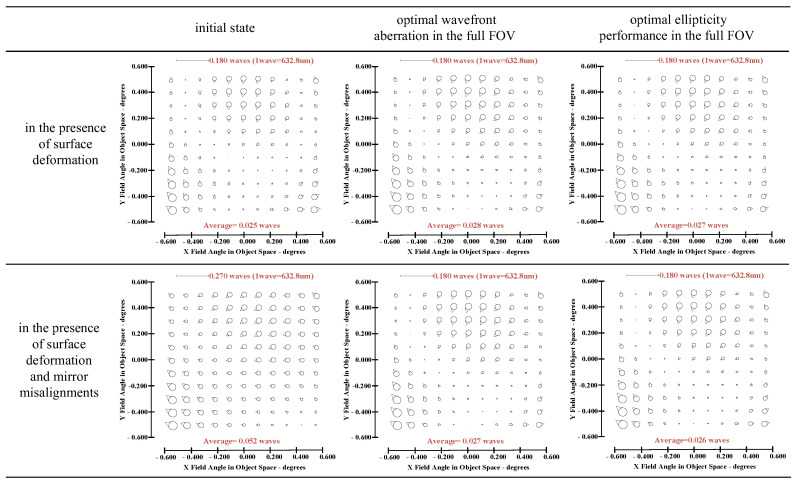
FFDs for coma (Z7/Z8).

**Table 1 sensors-23-04705-t001:** The typical values of ellipticity and RMS WFE in the different state.

Different State	Ellipticity		RMS WFE	
	Maximum Values	Mean Values	Maximum Values	Mean Values
Nominal state	0.0843	0.0234	0.0523λ	0.0256λ
In the presence of surface deformations	0.1152	0.0351	0.0778λ	0.0472λ
“Optimal wavefront aberration”	0.0857	0.0369	0.0647λ	0.0426λ
“Optimal ellipticity performance”	0.0670	0.0228	0.0825λ	0.0492λ

**Table 2 sensors-23-04705-t002:** The typical values of ellipticity and RMS WFE in the different state for complex working condition.

Different State	Ellipticity		RMS WFE	
	Maximum Values	Mean Values	Maximum Values	Mean Values
Nominal state	0.0843	0.0234	0.0523λ	0.0256λ
In the presence of complex working condition	0.1641	0.0854	0.6881λ	0.6448λ
“Optimal wavefront aberration”	0.0810	0.0374	0.0633λ	0.0430λ
“Optimal ellipticity performance”	0.0664	0.0233	0.0807λ	0.0486λ

## Data Availability

Not applicable.

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
