# Peer review of "Active Alignment of Large-Aperture Space Telescopes for Optimal Ellipticity Performance"

_sensors, 2023, doi:10.3390/s23104705_

Round 1

Reviewer 1 Report

The paper considers an active optical alignment strategy for achieving optimal ellipticity performance in large-aperture space telescopes. The proposed method utilizes nodal aberration theory and global optimization to determine the aberration field distribution for optimal full field-of-view ellipticity. The compensation DOFs of the secondary mirror and the folded flat mirror are then adjusted to achieve optimal ellipticity performance.

Simulations demonstrate that there is a certain difference between the state of optimal ellipticity performance and the state of optimal wavefront error. The mean ellipticity value corresponding to optimal ellipticity performance is lower than that of optimal wavefront performance, which indicates that achieving optimal ellipticity performance may not necessarily mean achieving optimal wavefront error.

The proposed method may provide a basis for the correction of ellipticity in complicated optical systems. The authors suggest that changing the goal of on-orbit alignment from optimal wavefront error to the goal of optimal ellipticity performance may be necessary if a higher ellipticity performance is required.

Artefacts from previous submissions to different journals like “(particle swarm optimization algorithm is used in this Letter)” in line 110 should be corrected.

The choice of particle swarm optimization algorithm in comparison with gradient methods should be explained.

IMG abbreviation in fig. 2 is not explained.

The materials (and their mechanical properties) used to manufacture mirrors of CSST should be provided.

Deformation value legend should be added to fig.4.

Table 2 do not correlate with the conclusions from figs. 9 and 10.

Figure 13 indicates not the astigmatism but coma.

The paper “Active alignment of large-aperture space telescopes for optimal ellipticity performance” could be published in Sensors after addressing the above comments.

Minor editing of English is welcome.

Author Response

Reviewer 1

(Please refer to the attachment for detailed content and pictures)

Comment 1):

Artefacts from previous submissions to different journals like “(particle swarm optimization algorithm is used in this Letter)” in line 110 should be corrected.

Author response: 

Thanks very much for this suggestion and we made corresponding revisions according to comments of the reviewer.

Author action:

In the revised manuscript, we change the contents “particle swarm optimization algorithm is used in this Letter” to “particle swarm optimization algorithm is used in this paper”

Comment 2):

The choice of particle swarm optimization algorithm in comparison with gradient methods should be explained.

Author response: 

We are really sorry that we did not make it clear for the issue, and the explanations are presented below:

  • Our research group widely applies particle swarm optimization algorithm and gradient algorithm, and relevant references are as follows:
  1. Particle Swarm Optimization

[1] Xin Qi , Guohao Ju , Shuyan Xu . Efficient solution to the stagnation problem of the particle swarm optimization algorithm for phase diversity[J]. Applied Optics, 2018, 57(11):2747.. SCI, IF: 1.97.

[2] Xiaoquan Bai, Guohao Ju, Boqian Xu, et al. Active alignment of space astronomical telescopes by matching arbitrary multi-field stellar image features [J]. Optics Express, 2021, 29(15), 24446-24465. SCI, IF: 3.529

……

  1. Gradient algorithm

[1] Zhang. Dong, Zhang. Xiaobin, Xu. Shuyan, Liu Nannan, Zhao Luoxin. Simplified Phase Diversity algorithm based on a first-order Taylor expansion [J]. Applied Optics, 2016, 55(28):7872-7877. SCI, IF: 1.65.

[2] Guo, L.; Ju, G.; Xu, B.; Bai, X.; Meng, Q.; Jiang, F.; Xu, S. Jitter-Robust Phase Retrieval Wavefront Sensing Algorithms. Sensors 2022, 22, 5584.

……

Gradient-based optimization algorithms can easily be trapped in a local minimum, for the searching direction of them mainly depends on the derivative information of the error metric.  At the same time, the objective function (in this paper) does not have an advantage in taking derivatives (ellipticity is calculated based on numerical methods). From this perspective, PSO algorithms are better, for they rely directly upon objective function values rather than derivative information.

  • Based on references [Xin Qi , Guohao Ju , Shuyan Xu . Efficient solution to the stagnation problem of the particle swarm optimization algorithm for phase diversity[J]. Applied Optics, 2018, 57(11):2747.. SCI, IF: 1.97.] and [Guo, L.; Ju, G.; Xu, B.; Bai, X.; Meng, Q.; Jiang, F.; Xu, S. Jitter-Robust Phase Retrieval Wavefront Sensing Algorithms. Sensors 2022, 22, 5584.] for simulation verification, the simulation results are shown in the following figure. We can see that PSO algorithm has significant advantages in correcting ellipticity compared to gradient algorithm (GA).
  • It's important to note that this paper proposes an active optical alignment strategy to achieve optimal ellipticity performance. Optimization algorithms are not the core of this article, they are only tools for solving objective functions.

Author action:

We add the following sentences after (particle swarm optimization algorithm is used in this paper) in section 3:

“Gradient-based optimization algorithms can easily be trapped in a local minimum, for the searching direction of them mainly depends on the derivative information of the error metric. At the same time, the objective function in this paper does not have an advantage in taking derivatives. From this perspective, PSO algorithms are better, for they rely directly upon objective function values rather than derivative information.[44]”

Add a reference: 44 Xin, Q.; Guohao, J.; Shuyan, X. Efficient solution to the stagnation problem of the particle swarm optimization algorithm for phase diversity. Applied Optics 2018, 57, 2747-2757.

Comment 3):

IMG abbreviation in fig. 2 is not explained.

Author response: 

We are sorry for the presence of such kind of mistakes, we made corresponding revisions according to comments of the reviewer.

Author action:

We made corresponding revisions in the revised manuscript: “including primary mirror (PM), secondary mirror (SM), tertiary mirror (TM) which is a freeform surface, a folding mirror (FM) which is used to correct image plane tilt and switch back-end astronomical measuring instruments and an image plane (IMG)”

Comment 4):

The materials (and their mechanical properties) used to manufacture mirrors of CSST should be provided.

Author response:

Thanks very much for providing these valuable suggestions, we made corresponding revisions according to comments of the reviewer.

On one hand, we made some revisions to the second paragraph of section 4: “which is used to correct image plane tilt and switch back-end astronomical measuring instruments and an image plane (IMG). The materials used to manufacture mirrors of CSST is SiC [ 47-50].

On the other hand, SiC is widely used in CIOMP. Every year, many conferences and papers discuss the characteristics of the material of SiC. At the same time, the surface deformation of the mirrors are closely related to the supporting structure, which is not the core content of this article, and a few short sentences cannot comprehensively introduce the characteristics of the mirrors. In order to give readers a deeper understanding, this article adds several references here.

Author action:

We made corresponding revisions in the revised manuscript:

we made some revisions to the second paragraph of section 4: “which is used to correct image plane tilt and switch back-end astronomical measuring instruments and a image plane (IMG). The materials used to manufacture mirrors of CSST is SiC [47-50].”

we provide references:

  1. Wang, K.; Dong, J. Structural design of Ф2 m-level large-diameter SiC reflector used in space remote sensor. Infrared and Laser Engineering 2017, 46.
  2. Wang, K.; Dong, J.; Wang, X.; Chi, C. Design of frame-type support structure for space-based rectangular convex mirror tested on the back. Optik - International Journal for Light and Electron Optics 2020, 212, 164673.
  3. Wang, K.; Dong, J.; Zhao, Y.; Chi, C.; Jiang, P.; Wang, X. Research on high performance support technology of space-based large aperture mirror. Optik: Zeitschrift fur Licht- und Elektronenoptik: = Journal for Light-and Electronoptic 2021, 226.
  4. Zhang, X.; Hu, H.; Wang, X.; Luo, X.; Zhang, G.; Zhao, W.; Wang, X.; Liu, Z.; Xiong, L.; Qi, E. Challenges and strategies in high-accuracy manufacturing of the world's largest SiC aspheric mirror. Light Sc,光:科学与应用(英文版) 2022, 11, 13.

Comment 5):

Deformation value legend should be added to fig.4.

Author response: 

We are sorry that we did not make it clear, we made corresponding revisions according to comments of the reviewer: Add a scale value to the right of each figure of mirror (including surface deformation).

Author action:

We made corresponding revisions in the revised manuscript. We will change Figure 4 to the following format:

(b)

(d)

(a)

(c)

Figure 4. The figure of each mirror (including surface deformation) considered in this paper for CSST: (a) figure of PM (b) figure of SM (c) figure of TM (d) figure of FM.

Comment 6):

Table 2 do not correlate with the conclusions from figs. 9 and 10.

Author response: 

We are sorry that we made wrong copy on Table 2, and we made corresponding corrections in the revised manuscript.

Author action:

We have corrected Table 2 and checked for any other data copy errors:

Table 2. The typical values of ellipticity and RMS WFE in the different state for complex working condition

Different state

ellipticity

RMS WFE

maximum values

mean values

maximum values

mean values

Nominal state

0.0843

0.0234

0.0523λ

0.0256λ

In the presence of complex working con-ditions

0.1641

0.0854

0.6881λ

0.6448λ

“optimal wavefront aberration”

0.0810

0.0374

0.0633λ

0.0430λ

“optimal ellipticity performance”

0.0664

0.0233

0.0807λ

0.0486λ

Comment 7):

Figure 13 indicates not the astigmatism but coma.

Author response: 

We are sorry for the presence of such kind of mistakes, we made corresponding revisions according to comments of the reviewer.

Author action:

We made corresponding revisions in the revised manuscript: “Figure 13. FFDs for coma (Z7/Z8).”

Reviewer 2 Report

This paper proposed an active optical alignment strategy to achieve optimal ellipticity performance and demonstrated the certain difference between the state of “optimal ellipticity performance” and the state of “optimal wavefront error. Based on the framework of nodal aberration theory (NAT), an active alignment method of large-aperture space telescopes for optimal ellipticity performance is proposed, the aberration field distribution which is corresponding to the optimal full field-of-view ellipticity is determined with global optimization and the degrees of freedom (DOFs) of the secondary mirror , the folded flat mirror is taken as the compensation DOFs to achieve the optimal ellipticity performance. 

There are two major questions about this paper to be considered:

1) The last paragraph on page 3 of the paper (line 110 of the full text) mentions the use of particle swarm optimization algorithm as a numerical optimization tool. However, no relevant instructions for using this algorithm were found in the paper. So the author should explain the parameter settings of the algorithm, and why the algorithm can reach the global optimum (paragraph 3, page 2, line 66 of the paper)?

2) At the end of Section 4.2, it is mentioned that the RMS WFE in the nominal state is 0.0256λ, and the corrected RMS WFE is 0.0492λ, those RMS WFE are still within the diffraction limit. Does this mean that the deformation of the surface in the simulation is too small? Is this in line with the actual scenario? In addition, can the correction result be understood as using the WFE of the system to compensate the ellipticity of the system's full field of view? The above questions also apply to Section 4.3.

There is no conmments on quality of English Language.

Author Response

Reviewer 2

(Please refer to the attachment for detailed content and pictures)

This paper proposed an active optical alignment strategy to achieve optimal ellipticity performance and demonstrated the certain difference between the state of “optimal ellipticity performance” and the state of “optimal wavefront error”. Based on the framework of nodal aberration theory (NAT), an active alignment method of large-aperture space telescopes for optimal ellipticity performance is proposed, the aberration field distribution which is corresponding to the optimal full field-of-view ellipticity is determined with global optimization and the degrees of freedom (DOFs) of the secondary mirror, the folded flat mirror is taken as the compensation DOFs to achieve the optimal ellipticity performance.

Comment 1): The last paragraph on page 3 of the paper (line 110 of the full text) mentions the use of particle swarm optimization algorithm as a numerical optimization tool. However, no relevant instructions for using this algorithm were found in the paper. So the author should explain the parameter settings of the algorithm, and why the algorithm can reach the global optimum (paragraph 3, page 2, line 66 of the paper)?

Author response:

  (1) The last paragraph on page 3 of the paper (line 110 of the full text) mentions the use of particle swarm optimization algorithm as a numerical optimization tool. However, no relevant instructions for using this algorithm were found in the paper.

We are really sorry that we did not make it clear for the issue, we should explain the parameter settings of the algorithm. We made corresponding revisions according to comments of the reviewer.

 (2) why the algorithm can reach the global optimum (paragraph 3, page 2, line 66 of the paper)?

the explanations are presented below:

  1. The particle swarm optimization (PSO) algorithm has simple structure, high convergence efficiency, and fast searching ability due to its parallel search mechanism.
  2. Gradient-based optimization algorithms can easily be trapped in a local minimum, for the searching direction of them mainly depends on the derivative information of the error metric. At the same time, the objective function (in this paper) does not have an advantage in taking derivatives (ellipticity is calculated based on numerical methods). From this perspective, PSO algorithms are better, for they rely directly upon objective function values rather than derivative information.

Author action:

First, we add the following sentences after (particle swarm optimization algorithm is used in this paper) in section 3:

“we set the learning factors c1=1.6, c2=2.65. The value of w is constant, w=0.5, T=0.001. The population size is 40 and the maximum number of iterations is 800.”

Secondly, we add the following sentences after (particle swarm optimization algorithm is used in this paper) in section 3:

“The particle swarm optimization (PSO) algorithm has simple structure, high convergence efficiency, and fast searching ability due to its parallel search mechanism. Gradient-based optimization algorithms can easily be trapped in a local minimum, for the searching direction of them mainly depends on the derivative information of the error metric. At the same time, the objective function in this paper does not have an advantage in taking derivatives. From this perspective, PSO algorithms are better, for they rely directly upon objective function values rather than derivative information [44].”

Add a reference: 44 Xin, Q.; Guohao, J.; Shuyan, X. Efficient solution to the stagnation problem of the particle swarm optimization algorithm for phase diversity. Applied Optics 2018, 57, 2747-2757.

Comment 2): At the end of Section 4.2, it is mentioned that the RMS WFE in the nominal state is 0.0256λ, and the corrected RMS WFE is 0.0492λ, those RMS WFE are still within the diffraction limit. Does this mean that the deformation of the surface in the simulation is too small? Is this in line with the actual scenario? In addition, can the correction result be understood as using the WFE of the system to compensate the ellipticity of the system's full field of view? The above questions also apply to Section 4.3.

Author response:

We are deeply impressed by the broad and profound knowledge of the reviewer. We are really grateful that our paper was reviewed by that who are the real expert in this field. It is a great honor for us to share our opinions.

 (1) At the end of Section 4.2, it is mentioned that the RMS WFE in the nominal state is 0.0256λ, and the corrected RMS WFE is 0.0492λ, those RMS WFE are still within the diffraction limit. Does this mean that the deformation of the surface in the simulation is too small?

We have to say that the deformation of the surface in the simulation is a little small. However, in fact, there are many other factors that will influence the imaging quality, such as line of sight (LOS) jitter, perturbations due to long-term drift in mirror figure and positions, the accuracy of wavefront sensing and control, et. al. Therefore, when only considering those aberration induced by optical design and mirror figure error, we need to leave a certain margin for wavefront error.

In addition, can the correction result be understood as using the WFE of the system to compensate the ellipticity of the system's full field of view? The above questions also apply to Section 4.3.

To some extent, your understanding is correct. The mutual matching of astigmatism, coma and defocus leads to the distribution of ellipticity. During the process of correcting ellipticity, it may be at the cost of reducing the average RMS WFE of full field of the system, but it meets the error requirements of space telescopes in-orbit. At the same time, in the design process of some telescopes, the distribution of wavefront aberration and ellipticity is balanced. As mentioned in reference [4] in original manuscript.

Reviewer 3 Report

Ms. Ref. No.: sensors-2378407

Title: Active alignment of large-aperture space telescopes for optimal ellipticity performance 

Article type: Full Length Article

Reviewer's Comments:

(1)  What is the reason the authors are using CSST as an example? Is there any other examples that can be taken into consideration for comparison?

(2)  In my opinion, the last paragraph of section 1 is not required as the current write up is not for thesis.

(3)   I would suggest the authors to provide citation for equation (4), (5) and (6).

(4)  It was noticed that CSST has be taken as the guideline for the report of this research work. I would suggest the authors to highlight the reason why CSST has been taken as a reference. What is the advantages provided by this CSST when compared with other available optical system? 

(5)  The authors should provide comparison between the obtain result in this work when compared with other reported results.

Author Response

Reviewer 3

(Please refer to the attachment for detailed content and pictures)

Comment 1):

What is the reason the authors are using CSST as an example? Is there any other examples that can be taken into consideration for comparison?

Author response: 

Thanks very much for this suggestion.

  • One of the scientific objectives of CSST is to detect dark matter and dark energy. One of the detection methods is achieved by measuring the ellipticity. At the beginning of CSST design, the distribution of ellipticity and wavefront aberration was balanced. As mentioned in reference [4] in original manuscript.
  • Correcting ellipticity is one of the requirements for CSST in orbit work.
  • The structure of space telescopes is complex, and it is difficult to integrate and simulate them. The relevant data of space telescopes being studied in various countries, such as Euclid mission and WFIRST, has not yet been made public. The author of this article was fortunate enough to participate in the development of the CSST telescope and was able to obtain relevant optical design data, which has a certain degree of authenticity. Taking CSST as an example, the method proposed in this article has certain engineering significance.
  • We applied reasonable scaling and loading of the surface deformation data to other optical systems, such as SNAP, and achieved good correction results, as shown in the following figure. Because these data are not from the actual operating conditions of the telescope, they were not included in the article.

Comment 2):

In my opinion, the last paragraph of section 1 is not required as the current write up is not for thesis.

Author response: 

Thanks very much for this suggestion and we made corresponding revisions according to comments of the reviewer.

Author action:

We made corresponding revisions in the revised manuscript, we delete the last paragraph of section 1:

“This paper is organized as follows. Section 2 introduces the definition of ellipticity and establish the relationships between ellipticity, aberration fields and misalignments. Active alignment strategy used to achieve optimal ellipticity performance is proposed in Section 3. Simulations are presented in Section 4 to demonstrate the effectiveness of the proposed strategy. Some other discussions are presented in Section 5 concerning the pro-portion between different aberration types in the state of optimal ellipticity performance.”

Comment 3):

I would suggest the authors to provide citation for equation (4), (5) and (6).

Author response: 

Thanks very much for this suggestion and we made corresponding revisions according to comments of the reviewer. However, the citation for equation (4), (5) have clarified in the last paragraph of section 2 in original manuscript.

At the same time, what we need to clarify is that Equation (6), as the objective function, is proposed in this paper rather than cited.

Author action:

We made corresponding revisions in the revised manuscript, we provide our previous article after equation (4) as citation for equation (4) and (5):

  1. Bai, X.; Guohao, J.U.Ju, G; Ma, H.; Xu, B.; Wang, S.; Xu, S. Active alignment of space astronomical telescopes by matching arbitrary multi-field stellar image features. Optics Express 2021, 29.

The detailed derivation process is presented in the above article.

Comment 4):

It was noticed that CSST has be taken as the guideline for the report of this research work. I would suggest the authors to highlight the reason why CSST has been taken as a reference. What is the advantages provided by this CSST when compared with other available optical system?

Author response: 

Thanks very much for this suggestion and we made corresponding revisions according to comments of the reviewer.

  • One of the scientific objectives of CSST is to detect dark matter and dark energy. One of the detection methods is achieved by measuring the ellipticity. At the beginning of CSST design, the distribution of ellipticity and wavefront aberration was balanced. As mentioned in reference [4] in original manuscript.
  • Correcting ellipticity is one of the requirements for CSST in orbit work.
  • The structure of space telescopes is complex, and it is difficult to integrate and simulate them. The relevant data of space telescopes being studied in various countries, such as Euclid mission and WFIRST, has not yet been made public. The author of this article was fortunate enough to participate in the development of the CSST telescope and was able to obtain relevant data, which has a certain degree of authenticity. Taking CSST as an example, the method proposed in this article has certain engineering significance.

Author action:

We made corresponding revisions in the revised manuscript, we clarify the reason why CSST has been taken as a reference. Add some sentence in first paragraph of section 4.1:

“In this paper, the optical system of the Chinese Space Station Telescope (CSST) will be used to demonstrate the effectiveness of the proposed approach (At the beginning of CSST design, the distribution of ellipticity and wavefront aberration was balanced. Correcting ellipticity is one of the requirements for CSST in orbit. The results of integrated simulation analysis (for CSST) are used for simulation analysis in this paper.)

Comment 5):

The authors should provide comparison between the obtain result in this work when compared with other reported results.

Author response: 

Thanks for your valuable comments. However, we cannot provide comparison between the obtain result in this work when compared with other reported results.

On one hand, nearly no other researchers specially study the correction of ellipticity. Traditional on-orbit active optical alignment of space telescopes usually takes “optimal wavefront aberration in the full field of view” as the correction goal, and most researches do not care the performance of ellipticity. Chinese Space Station Telescope (CSST) will be used to detect dark matter through weak lensing, and ellipticity performance affect the measurement of weak lensing. Therefore, correction of the ellipticity is a special question for CSST and other future space telescopes concerning the measurement of weak lensing.

On the other hand, in some papers, ellipticity of the PSF is used to correct the distribution of wave aberrations, as referenced in this article: [16] and [17]. The motivation of these papers is different from our manuscript.

Round 2

Reviewer 2 Report

the revised paper have meet the requirement of Sensors.